# Original Research Article

[†]Authors contributed equally

**Corresponding authors:**
Klaus Harter and Sven Nahnsen;
Emails: klaus.harter@zmbp.uni-tuebingen.de;
sven.nahnsen@uni-tuebingen.de;

**Associate Editor:**
Iain Johnston

# Nf-Root: A Best-Practice Pipeline for Deep-Learning-Based Analysis of Apoplastic pH in Microscopy Images of Developmental Zones in Plant Root Tissue

Julian Wanner[1,2,3,†], Luis Kuhn Cuellar[1,†] ⓘ, Luiselotte Rausch[4],
Kenneth W. Berendzen[4], Friederike Wanke[4], Gisela Gabernet[1],
Klaus Harter[4] and Sven Nahnsen[1]

[1]Quantitative Biology Center (QBiC), University of Tübingen, Tübingen, Germany; [2]Hasso Plattner Institute, University of Potsdam, Germany; [3]Finnish Institute for Molecular Medicine (FIMM), University of Helsinki, Helsinki, Finland; [4]Center for Plant Molecular Biology (ZMBP), University of Tübingen, Tübingen, Germany

## Abstract

Hormonal mechanisms associated with cell elongation play a vital role in the development and growth of plants. Here, we report Nextflow-root (nf-root), a novel best-practice pipeline for deep-learning-based analysis of fluorescence microscopy images of plant root tissue from A. thaliana. This bioinformatics pipeline performs automatic identification of developmental zones in root tissue images. This also includes apoplastic pH measurements, which is useful for modeling hormone signaling and cell physiological responses. We show that this nf-core standard-based pipeline successfully automates tissue zone segmentation and is both high-throughput and highly reproducible. In short, a deep-learning module deploys deterministically trained convolutional neural network models and augments the segmentation predictions with measures of prediction uncertainty and model interpretability, while aiming to facilitate result interpretation and verification by experienced plant biologists. We observed a high statistical similarity between the manually generated results and the output of the nf-root.

## 1. Introduction

One of the key mechanisms influencing the growth and morphogenesis of plants is cell elongation – a fast and irreversible increase in cellular size and volume. The kinetics and degree of cell elongation are regulated by several plant hormones as well as environmental signals and initially include cell modification through apoplastic pH change and alteration in turgor pressure. One of the key plant hormones influencing this process is brassinosteroids (BR). The role of BR, and more specifically that of brassinolide (BL) is an attractive target in a wide range of research fields, including pathogen defense, body patterning, response to abiotic stresses, and plant growth and development (Li et al., 2021; Witthöft & Harter, 2011; Yu et al., 2018; Caesar, Chen et al., 2011). BR are mainly perceived by the leucine-rich repeat (LRR) receptor-like kinase BRASSINOSTEROID-INSENSITIVE-1 (BRI1) (Z. Y. Wang et al., 2001). The binding of BL leads to the dissociation of BRI1-inhibitors BIK1, BKI1, and BIR3, enabling interaction and transphosphorylation with the co-receptor BAK1 followed by the activation of the major plasma membrane-localized proton pumping *Arabidopsis* H[+]-ATPases (AHAs) AHA1 and AHA2 (Caesar, Elgass, et al., 2011). This leads to the acidification of the apoplast and hyperpolarization of the plasma membrane, enabling cell wall loosening and eventually cell elongation (Caesar, Elgass, et al., 2011; Rayle & Cleland, 1970a; Zurek et al., 1994). Previous and recent experimental investigation of the BRI1 receptor complex in the plasma membrane (Caesar, Elgass, et al., 2011) led to a mathematical model of the fast-response pathway by assessing the apoplastic pH of *Arabidopsis thaliana* (*A. thaliana*) (Großeholz et al., 2022) using fluorescence microscopy (FM) and the ratiomeric fluorescent indicator 8-hydroxypyrene-1,3,6-trisulfonic acid trisodium salt (HPTS) (Barbez et al., 2017; Großeholz et al., 2022). However, despite the successful establishment

of a mathematical model relying on HPTS imaging, additional spectro-microscopic, plant-derived data are required to further expand and refine the model to capture additionally relevant parameters and processes, such as anisotropic growth in different root tissues and differential composition of BR-signaling complexes (Großeholz, 2019; Großeholz et al., 2020, 2022). Thus, to validate and further improve the fast-response pathway model, it is prudent to continuously generate new microscopic image datasets using the HPTS fluorescent indicator and, in the future, other pH-sensitive fluorophores, and analyze different tissue zones based on their morphology. In the case of the root tip, we choose the late elongation zone (LEZ), the early elongation zone (EEZ), and the meristematic zone (MZ) as suitable tissues. The annotated image data can then be subjected to image processing and statistical analysis of the derived values as required for the assessment. However, manual annotation of regions of interest (ROI) that correspond to tissue zones is an arduous and time-consuming task, which introduces a major bottleneck for the robust downstream quantitative analysis needed for predictive modeling.

Automatic segmentation of the above-mentioned morphological regions in microscopy images of root tissue can be formulated as a semantic segmentation problem, where class labels must be predicted for each pixel in the image. Significant advances in computer vision, through supervised machine learning, have made dense multi-class segmentation of biomedical images a tractable problem, especially since the introduction of deep convolutional models with encoder-decoder architectures, such as the U-Net model (Ronneberger et al., 2015) and its variants, e.g. U-Net++ (Zhou et al., 2018) and U-Net^2 (Qin et al., 2020). This approach allows biologists to automate microscopy image segmentation, by training predictive models using labeled datasets derived from manually annotated ROIs. Similar deep-learning models have been successfully applied to microscopy and medical image segmentation and in some cases providing human-level performance (Chlebus et al., 2018; Greenwald et al., 2022; Moebel et al., 2021). However, while deep convolutional neural networks achieve remarkable segmentation results, several reliability issues have been identified, in particular when applied to scientific data analysis (Belthangady & Royer, 2019). A significant shortcoming is non-reproducibility, both of deep-learning methods and the analysis pipelines that employ them, given that analysis reproducibility is a cornerstone of the scientific method (Collberg & Proebsting, 2016; Gundersen & Kjensmo, 2018; Haibe-Kains et al., 2020; Heil et al., 2021; Hutson, 2018). Another salient problem is the lack of a statistically sound measure to quantify the *uncertainty* of a model's prediction, as such a measure should be provided alongside segmentation predictions to permit appropriate interpretation of analysis results by experienced biologists and microscopists. This is particularly important since these methods are susceptible to biased and pathological predictions, especially when training data is limited (Amodei et al., 2016; Chen et al., 2018). Specifically, we refer to epistemic uncertainty, which relates to the limited amount of information a training dataset may provide about unseen and perhaps distal datapoints (Bernardo & Smith, 2009). While uncertainty quantification is already perceived as an essential property in machine learning methods (Kendall & Gal, 2017; Neal, 2012), it is still not often considered in biological data analysis. In this context, Gaussian processes (Grande et al., 2014; Rasmussen & Williams, 2005) are attractive probabilistic models, since they naturally model prediction uncertainty and have been shown to provide useful uncertainty quantification in experimental biology (Hie et al., 2020). Additionally, poor *interpretability* of

model output, that is, lack of information on why a segmentation prediction was produced for a particular input image, makes qualitative evaluation by experimental scientists a challenging task. Concretely, without a visual representation for which input features influence a prediction, it is problematic for biologists to assess whether the model detects the desired spatial features (e.g., features corresponding to the tissue morphologies) to perform predictions, or exploits artifacts in the dataset. Interpretability of deep convolutional models is a non-trivial task given the complexity of their prediction mechanisms and as a consequence, these models are often called "black boxes" (Alain & Bengio, 2016; Shwartz-Ziv & Tishby, 2017). However, recently developed methods, such as Guided Backpropagation (Springenberg et al., 2014) and Gradient-weighted Class Activation Mapping (Grad-CAM) (Selvaraju et al., 2017 can provide visual explanations of classification choices to enhance the transparency of deep convolutional models. Therefore, reproducibility, prediction uncertainty, and model interpretability are three characteristics that deep-learning algorithms require to be deployed within trustworthy and reliable tools for scientific data analysis. Hence, in order to apply deep convolutional models for image segmentation, the above-mentioned challenges need to be addressed to establish best-practice analysis methods for microscopy data.

Here we present a robust, best-practice pipeline to aid in the validation of the BR fast-response pathway model, which addresses the previously mentioned challenges. Importantly, we propose the use of the U-Net^2 model within the pipeline to alleviate the manual segmentation bottleneck, while implementing the means to calculate prediction uncertainty and interpret segmentation predictions. We built a highly reproducible image analysis pipeline using *Nextflow* (Di Tommaso et al., 2017) and *nf-core* tools (Ewels et al., 2020). This pipeline contains a module for the automatic segmentation of microscopy images, using a deterministically trained U-Net^2 model (i.e., bit-exact reproducible). To train this model, we created and made publicly available a supervised learning dataset for pH determination via FM, which we refer to as PHDFM. This dataset was created by acquiring confocal microscopy images of *A. thaliana* root tissue samples using the pH-sensitive HPTS indicator. Subsequently, these images were manually annotated with ROIs of relevant tissue zones and later used to generate multi-class, semantic segmentation masks. We then leveraged the *mlf-core* framework (Heumos et al., 2023) to deterministically train a PyTorch-based U-Net^2 model using the PHDFM dataset and built a *Nextflow*-compatible module for best-practice semantic segmentation with our deterministic model at its core. According to best practices, we used a well-established Bayesian approximation of the deep Gaussian processes (Gal & Ghahramani, 2016 to measure epistemic uncertainty from the U-Net^2 model and included this functionality in the segmentation module. Similarly, the module uses the Guided Grad-CAM algorithm (Selvaraju et al., 2017 to provide visual explanations for the segmentations predicted by the U-Net^2 model, aiming to provide means to interpret model results with biological knowledge. The segmentation module was then deployed within a *Nextflow* analysis pipeline, *nf-root*, which adopts *nf-core* reproducibility standards and implements the complete processing pipeline needed for high-throughput analysis. Finally, we applied this pipeline to analyze exemplary data and compared the results with those of an independent analysis, which was performed manually by experienced plant biologists. The comparison demonstrated a high similarity between the manually generated results and the output of the *nf-root*. Our results

suggest that this approach achieves near human-level segmentation performance, and results in a significant reduction in the time required to analyze this data from days to hours. Importantly, this method could be extended beyond the evaluation of this particular mathematical pathway model. The pipeline could be further developed to support associated research on root tissue that requires tissue-type-specific segmentations and statistics.

## 2. Results

### 2.1. The root tissue PHDFM dataset

For dataset creation, 601 FM images of *A. thaliana* root tissue were acquired with a confocal microscope (see *Materials and Methods* for details). The collected 2D images contain four channels, two for fluorescence signals and two for brightfield channels. From the four channels, the relative fluorescence ratio of the two fluorescent signals (attributed to protonated and deprotonated HPTS) can be easily obtained via previously reported image processing methods in Fiji (Barbez et al., 2017; Schindelin et al., 2012). Subsequently, regions of root tissue in all images were manually segmented into three different classes based on their morphology, namely the MZ, the LEZ, and the EEZ. Tissue zones were marked using labeled ROIs. Generally speaking, the zones were identified by noting that the MZ consists of cells with a Length/Width ratio < 1 while the Length/Width ratio of cells in the EEZ is between 1 and 2. The LEZ in turn consists of cells with a Length/Width ratio > 2. Consecutively two additional classes were added to label background and foreground (i.e. root tissue) pixels. The resulting ROI annotations were curated to generate multi-class pixel masks for semantic segmentation. For each image, a segmentation mask was generated to attribute each of the 512 x 512 pixels of the image to one of the following classes: background, root, MZ, EEZ, or LEZ. The resulting images with their corresponding segmentation masks were exported in OME-TIFF format (Goldberg et al., 2005; Linkert et al., 2010). This dataset was created using an OMERO server (Allan et al., 2012; Kuhn Cuellar et al., 2022) which acted as a scientific collaboration hub. The server allowed plant biologists to upload confocal microscopy images and annotate ROIs from their laboratory, while providing bioinformatic scientists with remote access to the data. This approach facilitated the collaboration needed to curate a dataset suitable to train a deep-learning model. Figure 1 provides a quantitative description of the PHDFM dataset, including the pixel label distribution for all classes (Supplementary Table S4). Example images of the PHDFM dataset are shown in Figure 1e.

### 2.2. Deterministic deep-learning for root tissue segmentation

We developed a package to segment different tissue types in confocal microscopy images of root tissue samples. This semantic segmentation package was built using *mlf-core* (Heumos et al., 2023), a framework that allows the implementation of fully deterministic, supervised machine learning applications using PyTorch. Our package implements state-of-the-art deep convolutional neural network models, specifically the U-Net, U-Net++, and U-Net^2 models (Qin et al., 2020; Ronneberger et al., 2015; Zhou et al., 2018), to segment brightfield images (bf-405nm) into regions of five classes: background, root tissue, EEZ, LEZ, and MZ. The U-Net^2 model outperformed the other two models with an average referred to as intersection over union (IoU) of 0.75, at the cost of a significantly larger number of trainable parameters (see

Supplementary Table S5). This package also provides functionality to automate hyperparameter optimization.

The training process (as depicted in Supplementary Figure S1) operates within a containerized software environment, using Docker containers and Conda environments to ensure a consistent and reproducible environment. The initial step of this process loads the PHDFM dataset, which is composed of 601 images and their corresponding segmentation masks, and splits it into three different subsets, a training set (80%) a validation set (10%), and a test set (10%). To increase the generalizability, operations for image data augmentation and perturbation were applied (a rotation of up to $10°$, shifting of up to 26 pixels, and scaling of up to 51 pixels). The above-mentioned operations were applied simultaneously in the data loading process with a probability of 50% each. For model training, a focal loss (Lin et al., 2020) was used as a loss function, and the Jaccard index, also IoU, was chosen as a metric to measure performance since these are well-established practices in the evaluation of semantic segmentation algorithms (Liu et al., 2021; Taha & Hanbury, 2015). Hyperparameter optimization uses a Bayesian and hyperband optimization strategy. For the hyperparameter optimization, the IoU of the EEZ, LEZ, and MZ classes was used to compare different models, as those are the classes of most importance for downstream data analysis (see *Best-practice Pipeline for Automated Ratiomeric Analysis* section). After hyperparameter optimization, our implementation of the U-Net^2 model (see *Materials and Methods*) obtained an average IoU of 0.75 in the validation dataset, and an average IoU of 0.75 in the test dataset.

To evaluate deterministic training (i.e. bit-exact reproducible), we repeated 10 times (n=10) the full training run (74 epochs), with our previously determined hyperparameters. As shown in Figure 2.b, we observed zero variance in the resulting model parameters (i.e. all trainable parameters, namely weights and biases) for the deterministic setup. Additionally, we measured IoU performance metrics for all training runs, including individual metrics for each segmentation class, and consistently observed no deviation using the deterministic setup (Figure 2.c). These evaluation results validate the reproducibility of our training workflow (Heumos et al., 2023). Visual inspection of individual image segmentations by an experienced plant biologist confirmed that model predictions provided qualitatively high segmentation results; sample segmentation predictions compared to the corresponding ground truths are shown in Figure 2.a (Supplementary Figure S3 shows examples of poor segmentation performance). Finally, a deterministically trained model is deployed for prediction as an additional Python module, which can then be used within a *Nextflow* pipeline (see *Best-practice Pipeline for Automated Ratiomeric Analysis*). Importantly, this module implements model interpretability functions to generate visual explanations of input feature importance using the Guided Grad-CAM algorithm (Selvaraju et al., 2017 and permits the calculation of prediction uncertainty maps using the Monte Carlo Dropout procedure (Gal & Ghahramani, 2016.

### 2.3. Prediction uncertainty in tissue segmentation

Segmentation predictions from the deep convolutional U-Net^2 model should be augmented with a measure of model uncertainty, aiming to provide plant biologists with useful information to inspect and interpret model predictions before conducting downstream statistics based on these results, allowing verification of the analysis results, e.g. for assessment of datapoint predictions that generate statistical outliers. Even though the deep convolutional

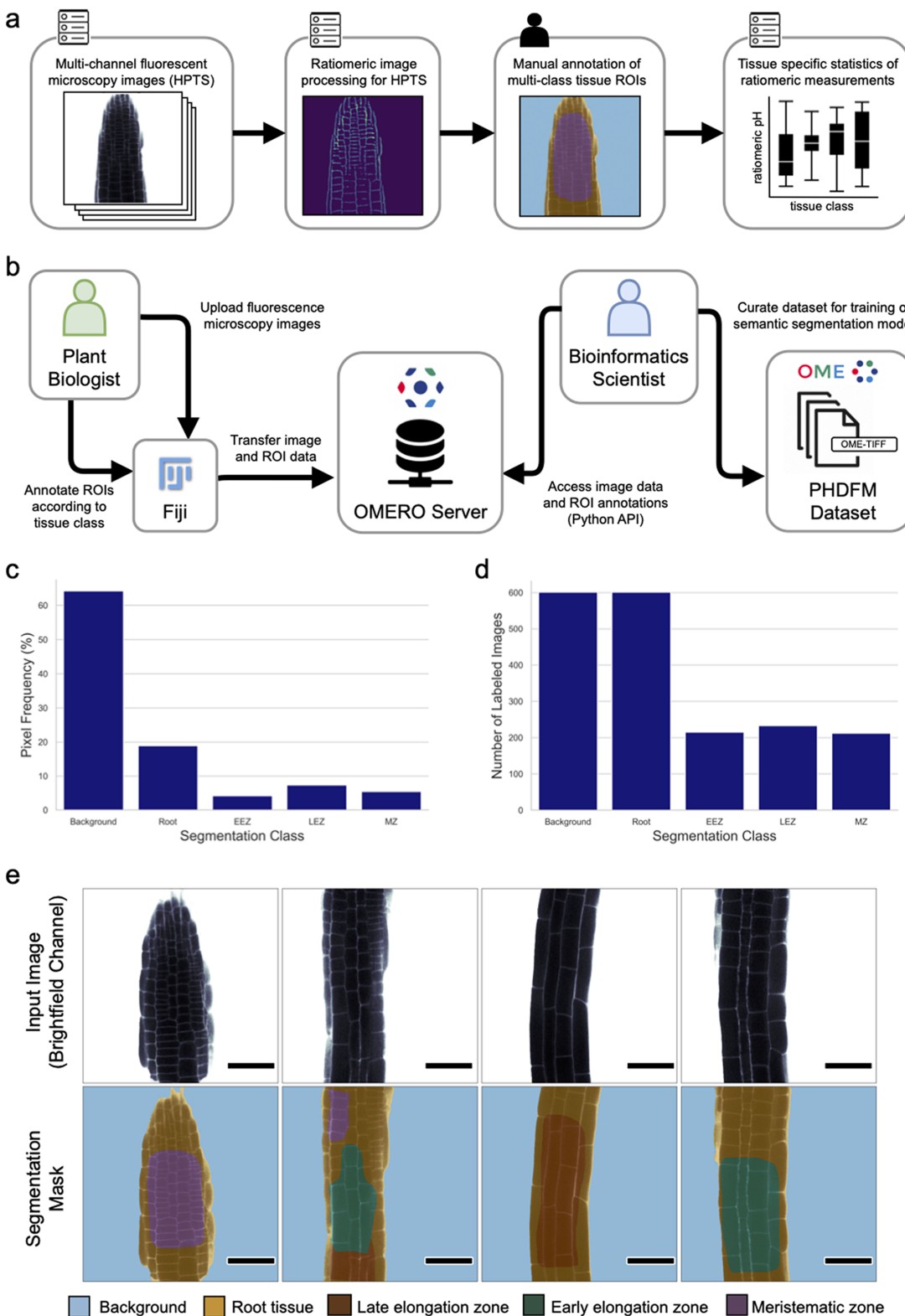

**Figure 1.** Ratiomeric pH analysis and the PHDFM dataset. (a) Diagram of the ratiomeric pH analysis for FM images from root tissue samples labeled with the HPTS marker. The input images typically have four channels (two brightfield and two fluorescence channels) to accommodate HPTS data. Manual annotation of ROIs is a time-consuming step, it creates the main bottleneck for large-scale data processing and precludes full automation of the complete analysis pipeline. (b) Diagram depicting the procedure used to create the dataset. An OMERO server was used as a remote collaboration hub between plant biologists and bioinformaticians to create a semantic segmentation dataset. (c) Frequency of annotated pixels per segmentation class in the dataset. (d) Distribution of the number of images containing pixel labels of each segmentation class. (e) Representative brightfield channel (bf-405nm) of FM images (top row) and corresponding labels of the PHDFM dataset (bottom row). Segmentation masks are depicted with color-coded tissue classes, showing five classes: background (blue), root tissue (yellow), LEZ (brown), EEZ (green), and MZ (purple). Scale bars = 53.14 μm.

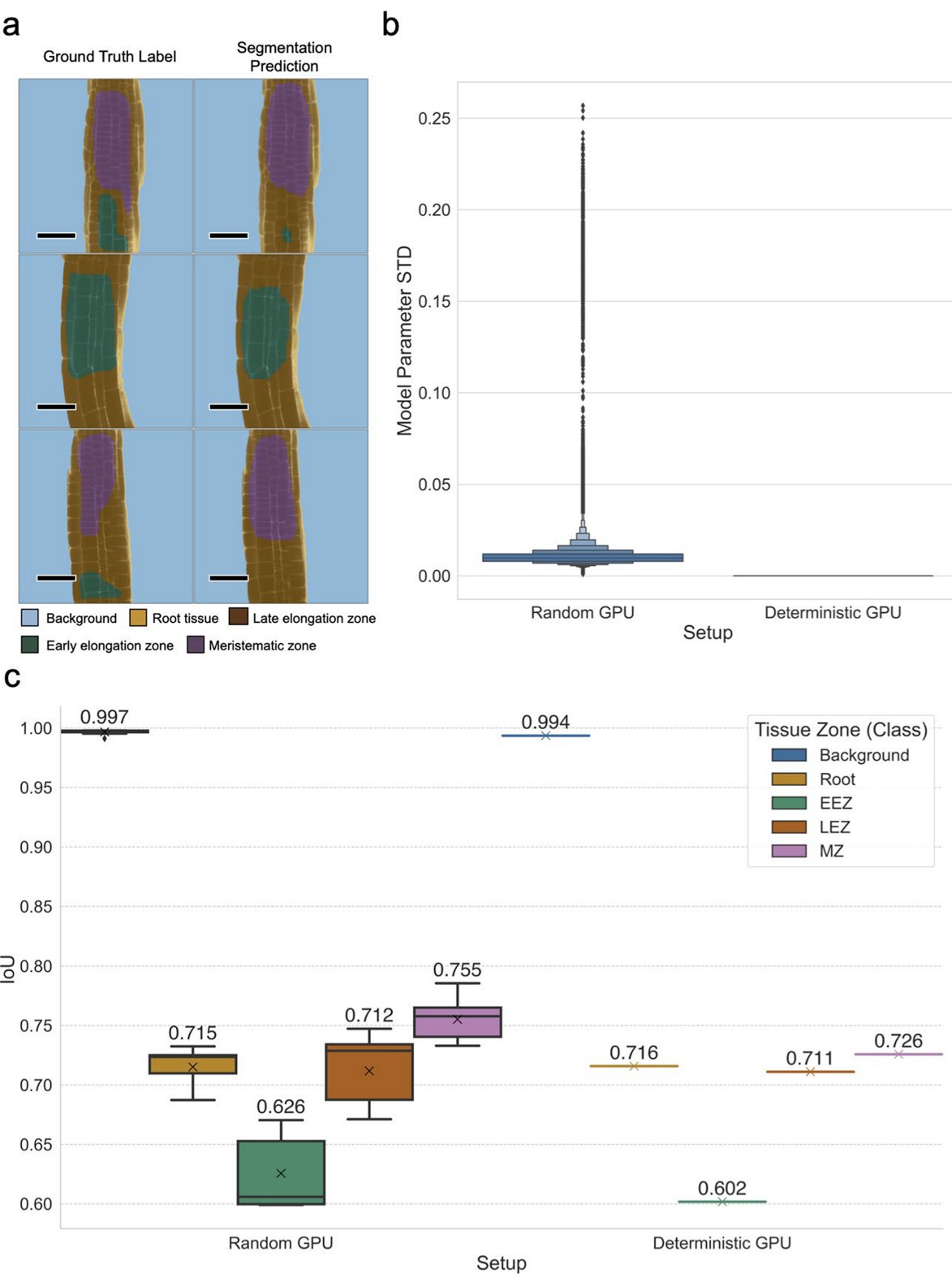

**Figure 2.** Qualitative and quantitative performance results of the U-Net^2 model, with an assessment of deterministic training. (a) Model predictions and ground truth show a high similarity, missing predictions are sometimes obtained when there are multiple labels in the ground truth. (b) Letter-value plot (Hofmann et al., 2017) of standard deviation values (STD) of U-Net^2 model parameters (weights and biases) across training runs (10 training runs per setup, n=10), the standard deviation of all 44.04 million trainable parameters was calculated for the Random (without random seed or deterministic settings) and Deterministic training setups (specified all random seeds and forced deterministic algorithms) (Heumos et al., 2023). (c) Boxplot of IoU performance on the test dataset (mean IoU of all images per class), after the training reproducibility test (n=10), this metric shows a large variance for all classes besides the background while using a non-deterministic setup and zero variance in all classes while using the deterministic setup, demonstrating full deterministic output of the training process. Scale bars = 53.14 μm.

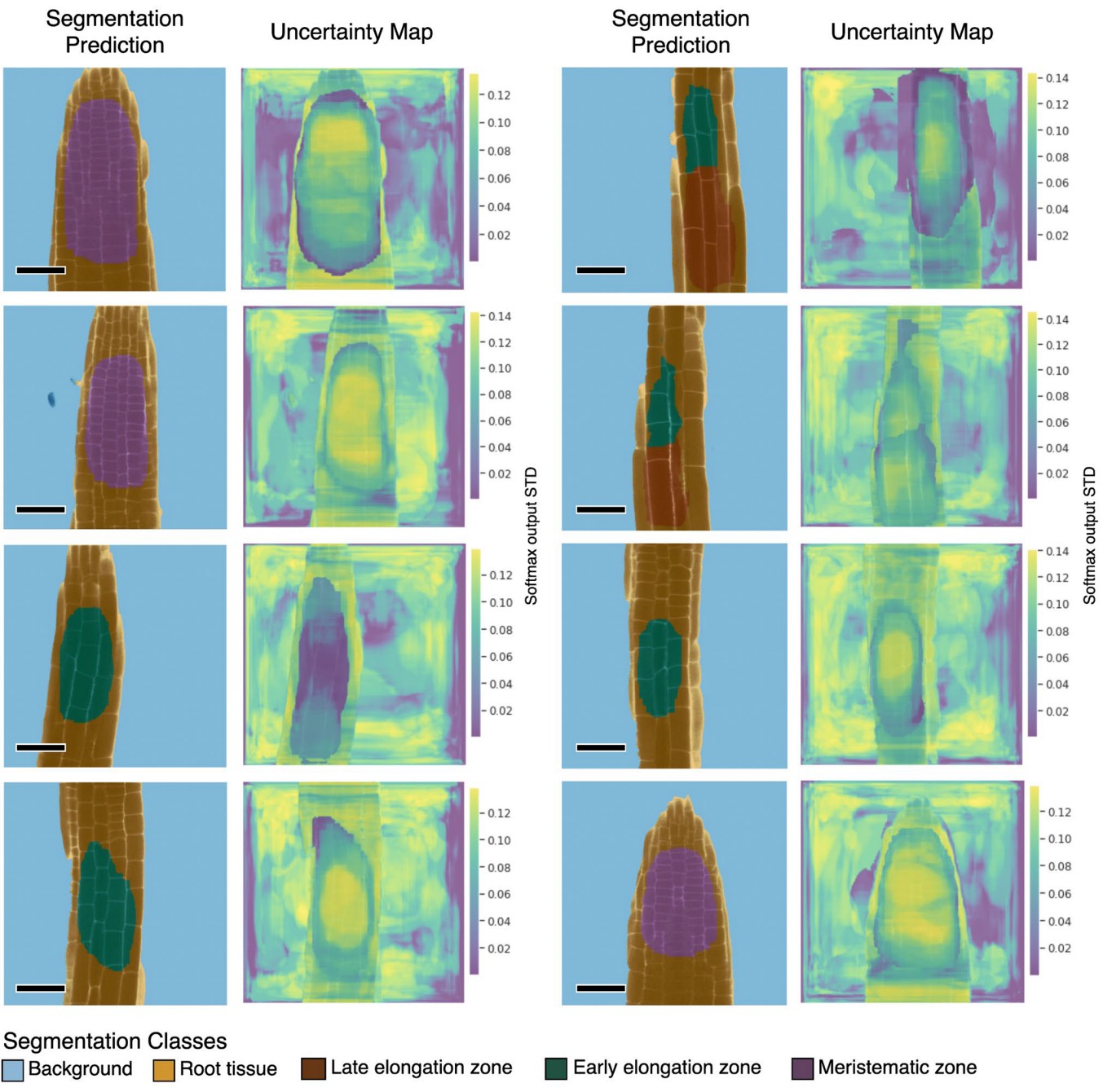

**Figure 3.** Representative samples of U-Net^2 segmentation predictions and their corresponding uncertainty maps. Predictions for images from a small unlabeled dataset. The uncertainty values are the pixel-wise, standard deviation values (STD) of the softmax output from the U-Net^2 model, as calculated using Monte Carlo Dropout. Uncertainty maps were calculated using Monte Carlo Dropout, with T=10 stochastic forward passes through the trained U-Net^2 model, and dropout applied before each convolutional layer in the model (dropout rate = 0.5). Pixels displayed in bright yellow relate to high uncertainty while pixels in dark blue represent low uncertainty. Scale bars = 53.14 μm.

U-Net^2 model provides high predictive performance, it is incapable of providing a measure of uncertainty on its dense, pixel-wise classification predictions. To ameliorate this issue, we employed a Bayesian approximation to cast our semantic segmentation model as a deep Gaussian process using the Monte Carlo Dropout procedure to measure prediction uncertainty (Damianou & Lawrence, 2013; Gal & Ghahramani, 2016; Kendall et al., 2015). This is a desirable approach since Bayesian probability theory provides a sound mathematical framework to represent uncertainty, and this method has been successfully applied to similar convolutional models for medical image segmentation (Wickstrøm et al., 2020).

In short, we applied dropout before each convolutional layer in the U-Net^2 model and used the Monte Carlo Dropout to sample sets of weight parameters from the trained model, aiming to approximate the predictive distribution. We set the dropout probability to 0.5 and the number of samples to 10 ($T = 10$). The output of Monte Carlo Dropout for an input image is an uncertainty map, where each pixel-wise uncertainty value is the standard deviation of the softmax outputs of the model, from T stochastic forward passes through the network. We calculated uncertainty maps for a reduced set of images (Figure 3). We observed regions of high uncertainty for image patches corresponding to all classes and often observed

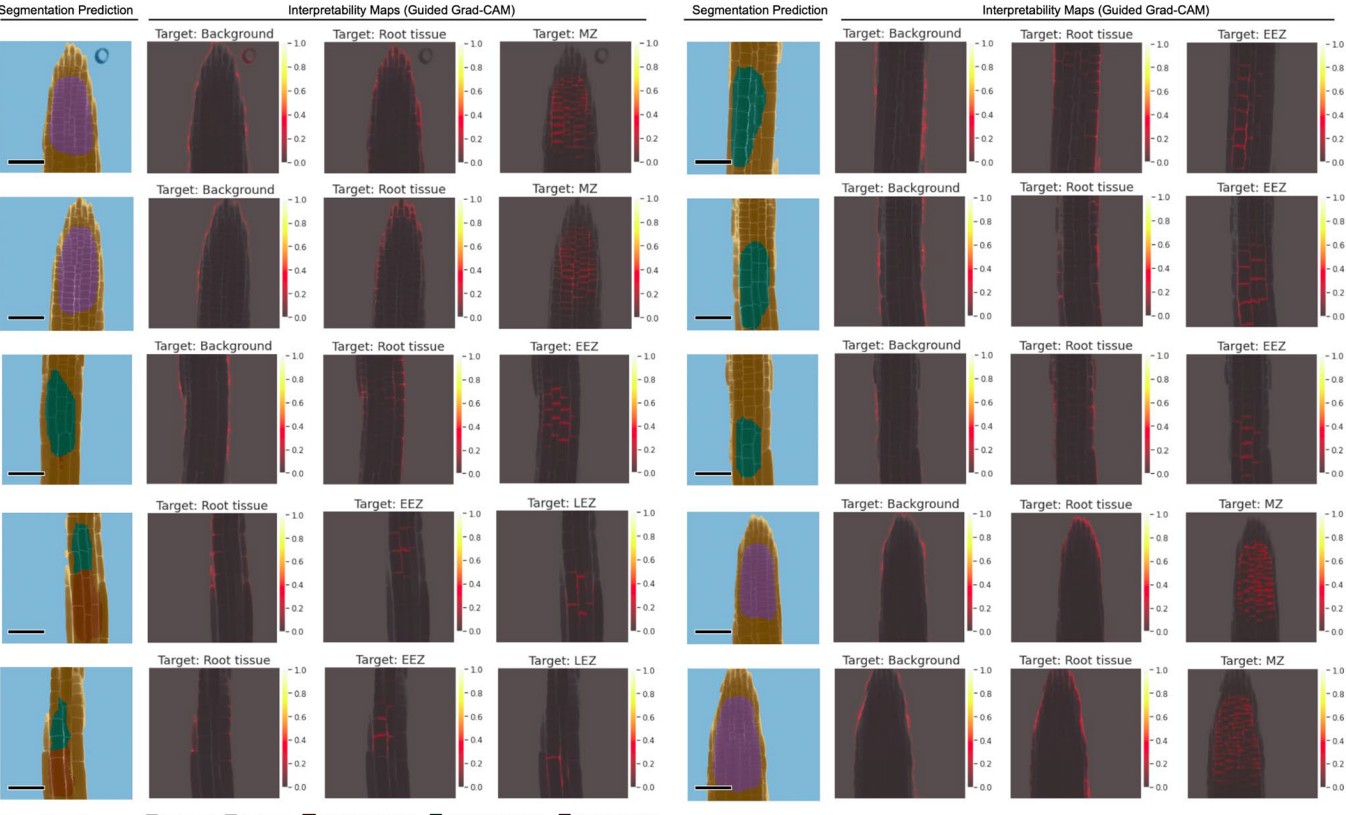

**Figure 4.** Representative samples of U-Net^2 segmentation predictions with their corresponding interpretability maps. Predictions for images from a small unlabeled dataset. Interpretability maps were calculated using Guided Grad-CAM, targeting the most relevant segmentation classes (Background, Root tissue, EEZ, LEZ, and MZ) for each image. Pixels in bright orange are highly important for the prediction of the target class (high importance score), and pixels in dark gray are associated with low feature importance for prediction. Scale bars = 53.14 μm.

high contrast of uncertainty values along the borders of regions of a different tissue type (i.e. "Root tissue", EEZ, LEZ, and MZ classes). Interestingly, within specialized tissue regions (i.e. EEZ, LEZ, and MZ classes), pixels along class borders exhibit lower uncertainty than those in inner zones, where predictions display noticeably higher uncertainty values (Figure 3).

### 2.4. Interpretability of the convolutional model for tissue segmentation

Interpretability, a sought-after property in deep convolutional models that are used for microscopy image analysis, refers to the ability to identify what input features drive the model to make a specific prediction. In this context, it is important to assess whether predictions are being made using information that agrees with plant biology knowledge, and thus build trust in the analysis. Considering that the convolutional U-Net^2 model on its own lacks the means to interpret its multi-class segmentation predictions, we apply the Guided Grad-CAM method (Selvaraju et al., 2017) to generate feature importance visualizations, which we refer to as interpretability maps. The output of Guided Grad-CAM fuses the results of the Guided Backpropagation (Springenberg et al., 2014) and Grad-CAM (Selvaraju et al., 2017) algorithms to provide highly class-discriminative, high-resolution visualization of spatial feature importance, in the input pixel-space. On one hand, the Guided Backpropagation algorithm (Springenberg et al., 2014) interprets the gradients of the neural network model with respect to the input image and provides clear and high-resolution visualizations

(i.e. fine-grained highlighting) of important spatial features in the input image (i.e. which pixels need to change the least to affect the prediction the most). On the other hand, Grad-CAM (Selvaraju et al., 2017) is a method to generate visual explanations for predictions from convolutional models, while it performs coarse-grained localization of prediction important regions within the input image, it is highly class-discriminative, localizing regions in input pixel-space that are only relevant to a specified target class.

We used Guided Grad-CAM to visualize interpretability maps from the U-Net^2 model for a small test set of images: Figure 4 shows representative examples. We targeted the most frequently predicted classes for each image to generate interpretability maps, where pixels in bright orange are considered highly important for the prediction of the target class, while pixels displayed in dark gray are associated with little importance for prediction (i.e. low feature importance). We observed that the model considered the border regions between the background and root tissue classes to differentiate and classify these large portions of the images. Importantly, the model correctly detects cell morphology within specialized tissue regions (i.e. EEZ, LEZ, and MZ classes), focusing heavily on spatial features derived from the structure and arrangement of cellular borders. The U-Net^2 model appropriately highlights cell borders for the EEZ and LEZ classes, since longitudinal cell lengths in these tissues are significantly larger than in MZ regions. Additionally, important features for the MZ tissue are likely to come from the structure of cellular borders of regions with a high density of significantly smaller cells.

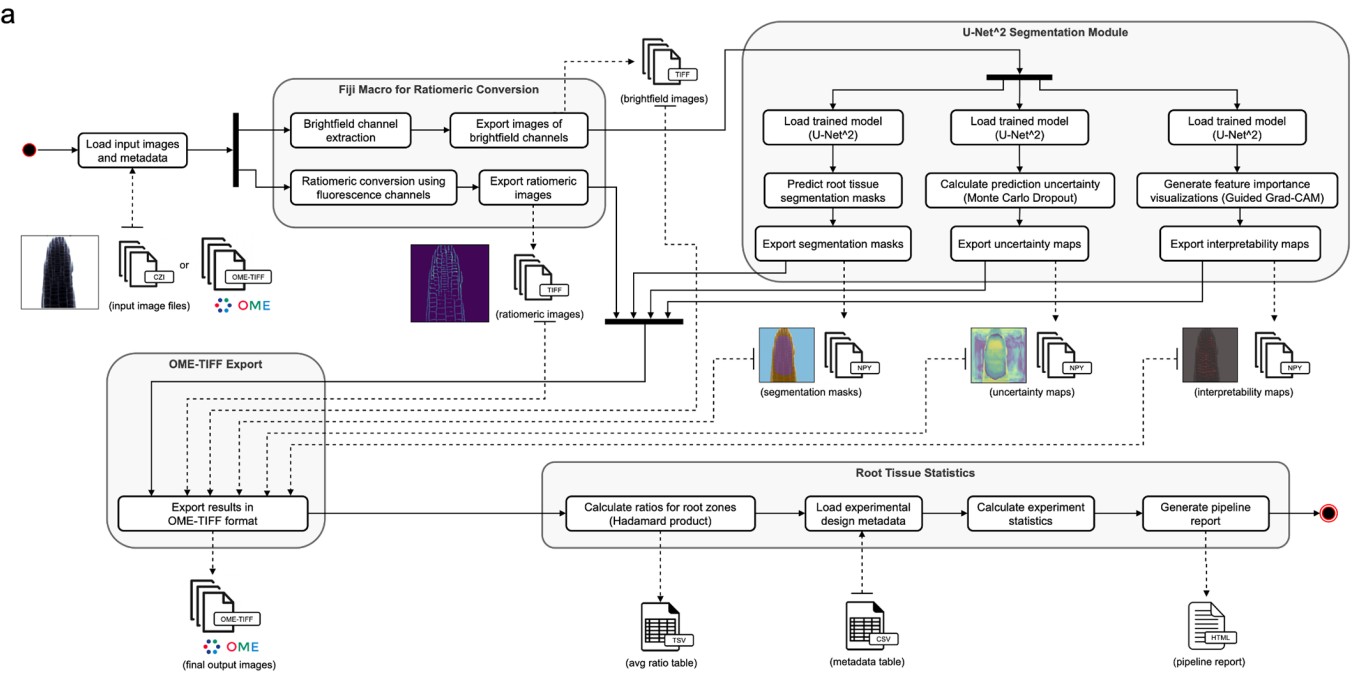

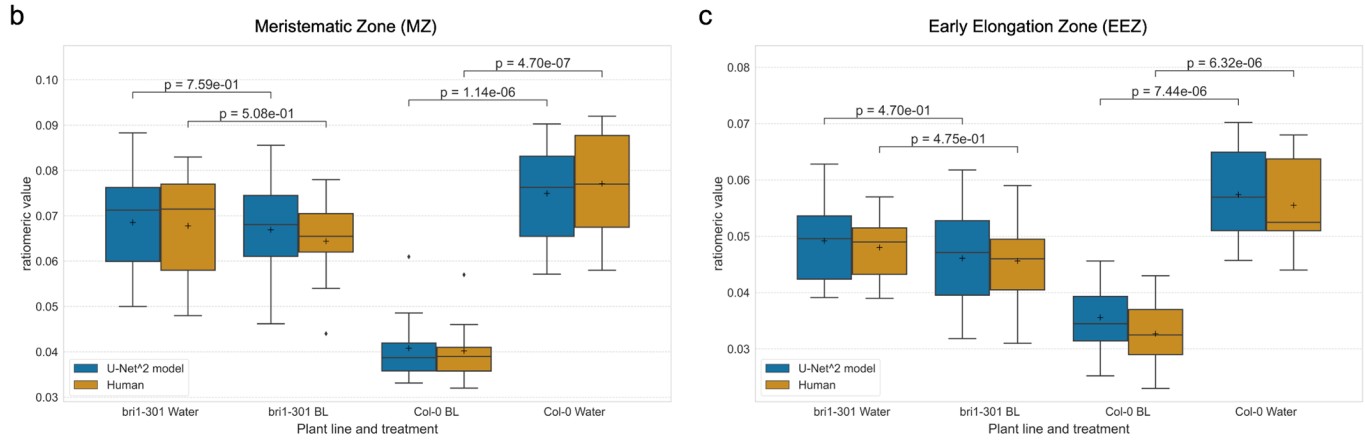

**Figure 5.** Quantitative evaluation of the nf-root pipeline. (a) Schematic of the nf-root workflow for the processing of A. thaliana image data. Inputs and outputs are denoted by dashed arrows, and surrounding rounded rectangles show docker containers for the respective task (N = 80). (b, c) Comparison of ratiomeric measurements between manual and automated analysis. Results from manual ROI annotation and data analysis by an independent plant biologist are shown in yellow (human), while automatic segmentation and analysis with our U-Net^2 model and the nf-root pipeline are shown in blue. Panel (b) shows ratio value statistics in the MZ, while (c) shows the corresponding values for EEZ, the most relevant zones for fast-response pathway analysis.

### 2.5. Best-practice pipeline for automated ratiomeric analysis

We integrated the U^2 best predictive model into *nf-root*, a *Nextflow*-based pipeline for the end-to-end analysis of FM images of root tissue, stained with a ratiomeric, pH-sensitive dye. This pipeline includes the analysis proposed by (Barbez et al., 2017) and automates tissue zone segmentation for downstream statistical analysis. The pipeline was built using *Nextflow* (DSL2) and *nf-core* tools, following the reproducibility standards of the *nf-core* community. The pipeline consists of 4 modules, as shown in Figure 5a.

The input of the pipeline is a metadata table (tabular file), and a set of images of 512x512 pixels in size, consisting of the four channels each: (1) fluorescence signal obtained by excitation at 405 nm, (2) brightfield image for excitation at 405 nm, (3) fluorescence signal obtained by excitation at 458 nm, (4) brightfield

image for excitation at 458 nm. The initial step of this pipeline is the Fiji (Schindelin et al.,2012) macro described in (Barbez et al., 2017) used to determine the ratio between the protonated state (absorption at 405 nm) and the unprotonated state (absorption at 458 nm) of HPTS, which makes it possible to assess the pH within the root tissue apoplast. The first brightfield channel is used as input for semantic segmentation using a Python package that deploys our previously trained U-Net^2 model (see *Deterministic Deep-Learning for Root Tissue Segmentation*). The model predictions and the pixel-wise ratio between protonated and unprotonated HPTS are then added as layers (i.e. as channels) to the original image, and the resulting multi-layer images are exported as OME-TIFF files. The OME-TIFF format was chosen to enhance data shareability and interoperability (Goldberg et al., 2005; Linkert et al., 2010), as it allows easy inspection of resulting images with Fiji and

Bio-Formats (Linkert et al., 2010), and facilitates the registration of this resulting imaging data into OMERO servers and robust data management (Allan et al., 2012). Subsequently, the average ratio value of a specific zone in one image is calculated as $avg(X \circ Y) = ratio$, where $X$ is the matrix containing ratios and $Y$ is the one-hot encoded label matrix of the specific zone. The provided metadata is used to conduct a Welch's t-test to compare different treatments, e.g. treatment with BL or water (control), of the plant root, and compared different mutant ecotypes with the Columbia-0 (Col-0) (Meinke & Scholl, 2003) wildtype of *A. thaliana*.

Finally, the pipeline generates a report (HTML-based document) presenting the above-mentioned analysis results, using a format tailored to the requirements of plant biologists. This report includes image segmentations and ratiometric images for each input image, allowing for visual inspection of results (see Supplementary Figure S2). Additionally, input feature importance (using Guided Grad-CAM (Selvaraju et al., 2017)) and prediction uncertainty (via Monte Carlo Dropout (Gal & Ghahramani, 2016)) are plotted for each segmentation prediction, with the aim of augmenting the information provided to the user, thereby allowing for better interpretation of the segmentation results.

We decided to compare the results of our pipeline against another, independent manual annotation and analysis, but this time include a never-before-seen plant variant (see *Materials and Methods*), with the aim of assessing if the pipeline results are indeed consistent with the gold standard procedure required to conduct this analysis. Biological samples of the *A. thaliana* mutant *bri1-301* were selected for comparison with Col-0 wildtype for microscopy data acquisition, since they are a prime example to assess the performance of the pipeline, given that this mutant has a defect in its BL pathway causing an inhibited root growth at room temperature, as it encodes the mutated kinase domain of the BRI1 protein (Xu et al., 2008).

Figure 5b and c depict a comparison between the ratio value distributions obtained using the *nf-root* pipeline against the corresponding distributions calculated using manual ROI determination by trained biologists. This comparison focuses on the LEZ and MZ classes since the abundance of *Bri1*-transcripts in the *A. thaliana* root is highest in the meristem and EEZ, while the transcript amount is comparatively low in the LEZ (Ma et al., 2020). These analyses show highly consistent results for corresponding experimental conditions. Importantly, significant BL-induced changes in the ratio values (and therefore in apoplastic pH) are only observed in the wildtype (*Col-0*), not in the *bri1-301* mutant. Furthermore, Welch's t-test statistics obtained using our pipeline are highly similar to those calculated using the equivalent procedure that employs manual ROI annotation. Additionally, the obtained ratios are similarly distributed with comparable means, medians, and quartiles for each tissue zone, treatment, and plant line. This indicates that the results produced by our pipeline can be used for this experimental setup and will greatly expand the amount of available data needed for testing the fast-response model.

## 3. Discussion

Here we report *nf-root*, a bioinformatic best-practice pipeline for apoplastic pH analysis of confocal microscopy images from tissues of the plant root tip. The goal of this approach was to gain extensive quantitative data sets for the refinement and extension of the mathematical model of the PM-located, BL/BRI1-dependent fast-response pathway regulating early processes that lead to

differential cell growth in the different zones of the root tip (Großeholz, 2019; Großeholz et al., 2022). This analysis pipeline was built using *Nextflow*, and *nf-core* tools (Ewels et al., 2020), making it easy to use, portable, and highly reproducible. Setting up the input for the pipeline is straightforward as it only requires a path to the image files and formatting of the metadata table (see Supplementary Table S6). *Nextflow* pipelines allow significant flexibility as they can be easily modified or extended by exchanging or adding processing modules. Moreover, this pipeline addresses the bottleneck of manual ROI annotation, by using a deep convolutional U-Net^2 model for automatic multi-class segmentation (five classes), providing high predictive performance and easy deployment via a containerized module. The implemented deep-learning module employs best practices for supervised machine learning to address major shortcomings of deep-learning methods in biological data analysis. In particular, we reach a high reproducibility standard by incorporating deterministic model training using the *mlf-core* framework (Heumos et al., 2023) and providing measures of uncertainty and interpretability for these segmentation predictions. Moreover, we compared the results of the *nf-root* pipeline against an independent data analysis procedure, which employs manual ROI annotation and data analysis, and observed that both methods yielded highly consistent and statistically similar results. Thus, providing qualitative and quantitative evidence that our approach achieves human-level performance.

Training of the U-Net models required the creation of the PHDFM dataset. The generation of datasets for semantic segmentation is a challenging task, as generating labeled datasets of significant size and label quality that allow training of deep convolutional models, requires plant biologists and microscopists to prepare biological samples, acquire microscopy data, and label ROIs at the large scale. Additionally, it requires bioinformatics scientists to process and curate this data (e.g. multi-channel images and ROI data) to generate a semantic segmentation dataset that can be directly used to train a deep convolutional model. Implementing microscopy data repositories with FAIR-oriented data management (Wilkinson et al., 2016) practices, for example by leveraging the OME data model (Linkert et al., 2010) and OMERO server (Allan et al., 2012), can facilitate the systematic augmentation of the PHDFM dataset with even more labeled data, that is, newly acquired confocal microscopy images with multi-class ROI annotations. These data repositories can then be used in strategies to improve the predictive performance of deep-learning models, by iteratively re-training and fine-tuning predictive models using constantly improving datasets, such as "human-in-the-loop" approaches that apply active learning methods (Greenwald et al., 2022; Pachitariu & Stringer, 2022). Similarly, sophisticated transfer learning (Jin et al., 2020; Shyam & Selvam, 2022; von Chamier et al., 2021; Wang et al., 2019) applications could be implemented in this setting, e.g. repurposing a U-Net model trained on the PHDFM dataset to aid in related tasks in root tissue biology, such as the segmentation of different tissue types and structures. This multidisciplinary, collaborative work centered around a data hub, an OMERO server, allows plant biologists to upload confocal microscopy data to the server, access those images and metadata via the web, and annotate ROIs using a web-based viewer or a Fiji plug-in. The OMERO server also allows bioinformaticians to access this microscopy data remotely, via direct download or programmatically, using Python and Java APIs, thus facilitating the curation of the dataset. We believe that similar approaches that facilitate collaboration between *in silico* and experimental

biology disciplines, will allow synergistic effects to accelerate the development of supervised machine learning methods and tools.

Here we advocate for the implementation of reliable deep-learning methods for semantic segmentation of microscopy data in life sciences. Specifically, such algorithms should augment neural network predictions with measures of *uncertainty* and *interpretability* to facilitate interpretation by experimental biologists. Here we observe that our U-Netˆ2 exhibits higher prediction uncertainty in transition regions between different tissue zones, while still successfully detecting EEZs, which are themselves transition zones in root tissue, and therefore can be challenging to identify even by trained plant biologists, as opposed to MEZ and LEZ regions. Moreover, interpretability maps indicate that regions located at tissue-type borders or transition areas contain important features that allow our model to detect MZs, EEZs, and LEZs. Thus, highlighting the importance of cell morphology gradients, i.e. the change in cell length/width ratios, which agrees with our biological understanding of tissue zone transitions. Interestingly, while these transition zones are of critical importance for tissue classification as shown by interpretability maps, these zones also exhibit lower prediction uncertainty, as indicated by uncertainty maps. While this result may be unexpected, it aligns with the heightened markedness of change in length/width ratio, which is particularly important for tissue classification. Moreover, a higher prediction certainty along the borders is especially convenient for the experimenter, since the border determination is usually more challenging compared to the classification of cells located in the center of MZ, EEZ or LEZ. Based on these observations, we believe further quantitative analysis of uncertainty and interpretability maps, perhaps using uncertainty thresholds, could aid in the identification of outlier input images or abnormal segmentation predictions.

The *nf-root* pipeline exemplifies the potential of machine learning methods to accelerate data analysis in life science, allowing high-throughput evaluation of experimental data. We followed open source and data guidelines (including training dataset, full hyperparameter documentation, and trained model), high reproducibility standards (e.g. deterministic training using *mlf-core*, *nf-core* compatible containerization), and incorporated prediction uncertainty and interpretability methods to allow for quality control, interpretation of segmentation predictions, and provide a measure of transparency for the deep convolutional neural network model. Importantly, our approach makes it possible to reproduce every part of the analysis workflow, including training and evaluation of the convolutional neural network models.

The power of our pipeline approach is reflected by the great similarity of the experimental data, evaluated by an experienced and well-trained researcher in a time-consuming process, and the *nf-root-generated* data, obtained in a much shorter time. For instance, *nf-root* is similarly able to determine the small difference between the LEZ and MZ in terms of the resting pH and the quantitatively differential acidification response of the two root tissues after BL application (represented as ratiometric value in Fig. 5b, c), as it was predicted by mathematical modeling (Großeholz et al., 2022). Moreover, the data also demonstrate that the apoplastic acidification after BL application requires kinase-active BRI1 in the LEZ and MZ as the kinase-inactive version of the receptor (BRI1-301) is not able to mediate an adequate cell physiological response, again supporting the prediction by our mathematical model (Großeholz et al., 2022).

*Summa summarum*, we introduce a development approach to build machine learning pipelines which adhere to best practices of scientific data analysis, including the principles of reproducibility, trustworthiness, and reliability. We believe that such practices are essential for the development of novel machine learning methodologies to analyze high-throughput data in life science, especially in combination with further biology-theoretical applications such as the mathematical modeling of cell physiological reactions presented here.

## 4. Materials and methods

### 4.1. Root tissue sample preparation

For pH-dependent FM using HPTS, *A. thaliana* plants were pre-grown on vertically placed 1/2-MS agar plates for 5 days. Two plant lines were used: the ecotype Col-0 (Meinke & Scholl, 2003), which served as the wildtype reference, and the *A. thaliana* BR mutant *bri1-301* (Col-0 background) (Lv et al., 2018; Zhang et al., 2018). Five seedlings were transferred to small Petri dishes containing 6 mL 1/2 MS medium with 60 µl 0.1M HPTS (end concentration 1 mM) and either 6 µL of dimethylsulfoxid (DMSO) as mock treatment or 6 µL 10 µM BL (end concentration 10nM). After 1 hour of incubation, the seedlings (embedded in medium) were carefully taken out of the Petri dish and carefully placed into microscopy imaging chambers upside down to allow imaging of the seedlings in close vicinity of the imaging chamber bottom (µ-Slide 2 Well Glass Bottom chambers, provided by ibidi GmbH from Gräfelfing, Germany). Air bubbles trapped between the sample and the surface as well as traces of the 1/2-MS medium were avoided as much as possible.

### 4.2. Microscopy data acquisition

Images of HPTS-stained root tissue were acquired with a Zeiss LSM880 confocal microscope. The acquired microscopy images have a size of 512 x 512 pixels. HPTS has two forms, a protonated and a deprotonated state. While both forms are fluorescent, they absorb at different wavelengths, 458 nm for the deprotonated and 405 nm for the protonated state. Thus, when acquiring root tissue images, samples were subsequently excited at these two wavelengths, and the corresponding fluorescence intensity was measured.

The confocal microscopy setup was the following: excitation lasers for 405 nm at 0.2 % of the maximum power, and for 458 nm at 100 % of the maximum power. The gain was set at 1200 AU. The detection range was set to 495 nm to 535 nm. Filters MBS458/514 and MBS-405 were used. The objective was water immersion with a magnification ratio of 40:1. Images of size 512×512 pixels were acquired. The target pixel size was set to 0.415133 µm.

The resulting fluorescence images contain four channels, divided into two brightfield channels and two fluorescence signal channels, according to the two wavelengths, which were consecutively acquired (458 nm and 405 nm). The channels are arranged in the following order: (1) fluorescence signal obtained by excitation at 405 nm, (2) brightfield channel for excitation at 405 nm, (3) fluorescence signal obtained by excitation at 458 nm. (4) brightfield channel for excitation at 458 nm.

### 4.3. Annotation of ROI

The image dataset was registered into an OMERO server (Allan et al., 2012). The Fiji application (Schindelin et al., 2012), with the OMERO and Bio-Formats (Linkert et al., 2010) plug-ins, was used to manually annotate ROIs and import them into an

OMERO server instance. Class-labeled ROIs were annotated by visual inspection of the brightfield channel for excitation at 405 nm (bf-405nm), as this channel was preferred by plant biologists for this task.

## 4.4. Dataset processing

The image data and annotated ROIs were fetched from the OMERO server and processed to generate semantic segmentation masks for all images. These masks define the pixel-wise class assignment. The segmentation masks classify pixels into one of the following 5 labels with the corresponding numeric identifiers: background (0), root tissue (1), EEZ (2), LEZ (3), and MZ (4).

## 4.5. Ratiomeric image processing

The ratiomeric values calculated by the Fiji macro (Barbez et al., 2017; Schindelin et al., 2012) depict the relative fluorescence of the HPTS dye, sequentially excited at two different wavelengths, 458 nm for the deprotonated form, and 405 nm for the protonated form. The relative abundance of protonated and deprotonated HPTS is then reflected by the pixel-wise, 458/405 intensity ratio.

## 4.6. Implementation of root tissue segmentation

The Optuna framework (Akiba et al., 2019) was used for hyperparameter optimization. Optimal hyperparameters are used as default values in the training module. The documentation of the hyperparameter can be found in the mlf-core-based training module (https://github.com/qbic-pipelines/root-tissue-segmentation-core/blob/master/docs/usage.rst), which reflects the default settings (https://github.com/qbic-pipelines/root-tissue-segmentation-core/blob/master/MLproject), software and hardware information are also available in the module (https://github.com/qbic-pipelines/root-tissue-segmentation-core). We used version 1.0.1 of the segmentation training module.

All semantic segmentation models were implemented in a PyTorch-based mlf-core (Heumos et al., 2023; Paszke et al., 2019) project, using the brightfield channel for sample excitation at 405 nm (bf-405nm) as the input image, and segmentation masks with 5 classes (background, root tissue, EEZ, LEZ, MZ). The U-Net^2 model architecture was implemented as previously described (Qin et al., 2020). Aside from changes in input channels (one brightfield channel) and the number of pixel classes (5 classes), the model structure was preserved, including kernel sizes for convolution, number, and structure of layers, operations, and residual U-blocks. The trained U-Net^2 model can be found here: https://zenodo.org/record/6937290

During the training process, operations for image data augmentation and perturbation were applied. Specifically, a rotation of up to $10°$, shifting of up to 26 pixels, and scaling of up to 51 pixels. We selected these transformation parameters based on the technical specifications of the microscopy instruments, the experimental variation we observed during sample preparation, and the data acquisition parameters. More specifically, we estimated these transformation parameters based on qualitative, visual assessment of the brightfield images, and the acquisition parameters specific to the dataset at hand (e.g., image size, target spatial resolution, physical pixel size, etc.). In short, informed selection follows the empirical investigation of parameters.

We calculate the Jaccard index, also referred to as IoU, to evaluate the predictive performance of the trained models. For each class $c$ in an image, the IoU metric can be expressed using logical operators as in (Wickstrøm et al., 2020):

$$IoU(c) = \frac{\sum_i (\widehat{y}_i == c \wedge y_i == c)}{\sum_i (\widehat{y}_i == c \vee y_i == c)}$$

Where the segmentation masks $\widehat{y}$ and $y$ denote the model prediction and ground truth, respectively, for each pixel-wise prediction $i$ in an image input.

## 4.7. Implementation of uncertainty and interpretability functionality

To calculate the uncertainty of segmentation prediction we used the Monte Carlo Dropout procedure. Using dropout as a Bayesian approximation to deep Gaussian processes (Gal & Ghahramani, 2016), the predictive distribution of our segmentation model can be approximated using Monte Carlo integration, as expressed by (Wickstrøm et al., 2020):

$$p(y_* \mid x_*, D) \approx \frac{1}{T} \sum_{t=1}^{T} softmax\left(f_{W_t^*}(x_*)\right)$$

Where the dataset $D$ is composed of a set of pairs, from input images and their corresponding label masks, and $x_*$ and $y_*$ are a new pair of input images and label masks. Here $W_t^*$ refers to the stochastically sampled weights of the model for sample $t$, and $f_W(x_*)$ is the output of the model (i.e. a forward pass). To create the uncertainty maps, we enable dropout, and calculate the standard deviation of the softmax outputs of the U-Net^2 model, after $T$ forward passes.

In the context of prediction uncertainty via the Monte Carlo Dropout procedure, the prior distributions for the model parameters (i.e. weights of the neural network) are not explicitly defined, as in conventional Bayesian approaches. The Monte Carlo Dropout method interprets dropout as a practical approximation to Bayesian inference, with dropout rates influencing the variance of the prior distribution over the weights. Monte Carlo Dropout implicitly regularizes models by acting as an unspecific prior, favoring simplicity to prevent overfitting. The form of the implicit prior depends on how dropout is implemented, and used within the neural network architecture. By enabling dropout during inference, running inference multiple times, and aggregating the outcomes, this method approximates the posterior distribution of the weights, offering an efficient method to estimate prediction uncertainty in deep learning, without detailing priors for all model parameters.

The functionality to apply Guided Grad-CAM to the U-Net^2 segmentation model was implemented using the Captum library (Kokhlikyan et al., 2020). We compute attribution by summing the output logits of each channel, corresponding to a total score for each segmentation class, and attributing with respect to this score for a particular class. We sum only the scores corresponding to pixels that are predicted to be of the target class (i.e. when the argmax output equals the target class) and attribute with respect to this sum. We define a wrapper function, as described in the captum documentation (https://captum.ai/tutorials/Segmentation_Interpret).

The implementation of Monte Carlo Dropout and Guided Grad-CAM can be found in the prediction module (https://github.com/qbic-pipelines/rts-prediction-package).

## 4.8. Implementation of the analysis pipeline

The analysis pipeline was implemented in *Nextflow* (Di Tommaso et al., 2017), using DSL2 (https://www.nextflow.io/docs/latest/

dsl2.html). The *mlf-core* segmentation prediction module (https://github.com/qbic-pipelines/rts-prediction-package), the *nf-root* pipeline created using *nf-core* tools v 2.2.dev0 (https://github.com/qbic-pipelines/root-tissue-analysis), and the test dataset for the pipeline (https://zenodo.org/record/5949352/) are publicly available online. We used version 1.0.7 (Mark-1.0.7) of the segmentation prediction module and version 1.0.1 of the *nf-root* pipeline. This pipeline takes as input FM image files in *.czi* (Zeiss CZI) (https://docs.openmicroscopy.org/bio-formats/6.11.1/formats/zeiss-czi.html) or in *.ome.tif* (OME-TIFF) (Linkert et al., 2010) file format. The pipeline outputs all imaging data (e.g. segmentation masks, uncertainty, and interpretability maps) in OME-TIFF format.

### 4.9. Comparing nf-root with manual annotation and analysis

The dataset of raw confocal microscopy images used for performance comparison between the nf-root pipeline and an analysis performed by an experienced biological experimenter was generated as follows. Images were acquired as described in sections "Root tissue sample preparation" and "Microscopy data acquisition". Generation of ratiometric images using the Fiji-macro published by (Barbez et al., 2017), annotation of ROIs in Fiji and statistical evaluation by Welch's t-test, and inspection of key statistical parameters were performed independently by the biological experimenter (without involvement of any nf-root process). Importantly, this analysis was performed manually, and independently from the analysis based on the nf-root pipeline.

### Acknowledgments

This work was supported by the BMBF-funded de.NBI Cloud within the German Network for Bioinformatics Infrastructure (de.NBI) (031A532B, 031A533A, 031A533B, 031A534A, 031A535A, 031A537A, 031A537B, 031A537C, 031A537D, 031A538A), the DFG-funded CRC 1101 project D02 to K.H., and the DFG instrumental funds INST 37/819-1 FUGG and INST 37/965-1 FUGG. SN acknowledges support by DataPLANT, funded by the German Research Foundation (DFG) within the framework of the NFDI [442077441].

**Competing interest.** The authors declare no competing interests.

**Author contributions.** JW and LKC wrote the pipeline and modules, performed analyses, curated the PHDFM dataset, and generated the figures. KWB and LR validated and analyzed experimental data. FW acquired microscopy imaging data and performed manual segmentation for the PHDFM dataset. SN, GG, and KH supervised the work. All authors wrote the manuscript, provided critical feedback, and helped shape the research, analysis, and manuscript.

**Data availability.** The PHDFM dataset is available at https://zenodo.org/record/5841376/.

**Supplementary material.** The supplementary material for this article can be found at http://doi.org/10.1017/qpb.2024.11.

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
