## [Editor Report]

Thanks for your submission to QPB. It’s now been reviewed by two independent reviewer sets -- one reviewer and one pair of reviewers working together, and I’ve taken a look too. To me this feels like a careful and deep exploration of a potentially useful tool and with some changes to the manuscript and, most importantly, the software implementation, it could be a valuable tool.

However, one reviewer raises the important concern that the software as it stands doesn’t work. Given the admirable focus on computational good practise and reproducibility, this is a critical fix. I’m therefore recommending major revisions -- though hopefully fixing these bugs won’t require substantial effort.

The reviewers also raise some points at the manuscript level that should be addressed in a round of revision. I also had the below comments (which may overlap).

The abstract in the ms and in the online system are different, perhaps due to an imposed word limit? If you are satisfied with the version that obeys this limit please include that in the manuscript.

The first intro paragraph sets up the knowledge gap in a slightly roundabout way which perhaps sells short the general applicability of this approach. It will presumably be of use not just for further development of a particular mathematical model, but also for lots of associated research.

Several citations are oddly formatted e.g. Chen & Johansson, n.d.; Grande et al., 22--24 Jun 2014. If the references can be systematised it will make future steps smoother.

Why were the particular limits for scaling and shifting (51 and 26 pixels) chosen? Won’t this behaviour depend on the resolution / size of the images?

Fig 2b -- it’s hard to interpret these STD values without an idea of the typical parameter scale. Are parameters, for example, typically on the unit interval? Similarly with the scale of the colourmaps in Fig 3. What does an uncertainty of 0.1 actually correspond to? 10% realtive error perhaps?

What are the priors in the Bayesian analysis, and what influence do they have?

In Fig 5 would it be possible to have a side-by-side comparison of a “good” result (pipeline matches human) and a “bad” result (mismatch) so the reader has an idea of the scale of differences involved?

---

## [Editor Report]

Thanks for your revised submission, which I and the reviewers agree has fixed several bugs and cleared up most questions. The software now appears to be functional, and useful! I am happy to recommend acceptance contingent on a couple of remaining additions which shouldn’t take more than a few minutes to include. In addition, one reviewer has some further suggestions for the code that could be included in future updates.

A few questions from the previous round didn’t lead to edits in the manuscript. The response letter gave a good explanation for the particular scaling and shifting used -- please put this explanation in the manuscript (informed selection following empirical investigation is a perfectly valid answer).

In the last round I requested a comparison between an image where the pipeline performs well and one where it performs poorly. As the authors respond, the former case is in Fig 2a. But I (and a reviewer) would still like an example of less-good performance. Certainly the statistics are presented in Fig 5b-c (as in the response) but the point of this question is to help the reader understand what these values actually look like in a real image. Including an example image or two, like Fig 2a but for a poor-quality results, would immediately do this -- it could be a supplementary image if you don’t want to break the flow of the existing ms.

All the best,

Iain

---

## [Editor Report]

Great -- I think this important detail and context is now suitably included, reviewer comments have been addressed, and am happy to recommend acceptance. Thanks for working with QPB!